# Characteristics of Fragmented Aurora-like Emissions (FAEs) observed on Svalbard

Joshua Dreyer[1,2], Noora Partamies[3,4], Daniel Whiter[5], Pål G. Ellingsen[6], Lisa Baddeley[3,4], and Stephan C. Buchert[1]

[1]Swedish Institute of Space Physics (IRF), Space Plasma Physics group, Uppsala, Sweden
[2]Uppsala University, Department of Physics and Astronomy, Uppsala, Sweden
[3]The University Centre in Svalbard, Longyearbyen, Norway
[4]Birkeland Centre for Space Science, Bergen, Norway
[5]University of Southampton, Department of Physics and Astronomy, Southampton, UK
[6]UiT The Arctic University of Norway, Department of Electrical Engineering, Narvik, Norway

**Correspondence:** Joshua Dreyer (joshua.dreyer@irfu.se)

**Abstract.** This study analyses the observations of a new type of small-scale aurora-like feature, which is further referred to as Fragmented Aurora-like Emission(s) (FAEs). An all-sky camera captured these FAEs on three separate occasions in 2015 and 2017 at the Kjell-Henriksen Observatory near the arctic town of Longyearyben, Svalbard. A total of 305 FAE candidates were identified. They seem to appear in two categories - randomly occurring individual FAEs and wave-like structures with regular spacing between FAEs alongside auroral arcs. FAEs show horizontal sizes typically below 20 km, a lack of field-aligned emission extent and short lifetimes of less than a minute. Emissions were observed at the 557.7 nm line of atomic oxygen and at 673.0 nm ($N_2$, first positive band system), but not at the 427.8 nm emission of $N_2^+$ or the 777.4 nm line of atomic oxygen. This suggests an upper limit to the energy that can be produced by the generating mechanism. Their lack of field-aligned extent indicates a different generation mechanism than for aurora, which is caused by particle precipitation. Instead, these FAEs could be the result of excitation by thermal ionospheric electrons. FAE observations are seemingly accompanied by elevated electron temperatures between 110–120 km and increased ion temperatures at F-region altitudes. One possible explanation for this are Farley-Buneman instabilities of strong local currents. In the present study we provide an overview of the observations and discuss their characteristics as well as potential generation mechanisms.

## 1 Introduction

Aurora as a phenomenon has been studied extensively over the past century, and mesoscale auroral forms like arcs are generally rather well-understood. Some open questions remain though, such as the intricacies of sudden changes in morphology and the drivers behind dynamic auroral processes (Karlsson et al., 2020). Small-scale features on the other hand are much less well-known and new features are still being found, for example transient phenomena such as Lumikots (McKay et al., 2019).

Auroral emission is dependent on the atmospheric composition, which varies with altitude. The same wavelengths that are typically observed with aurora can also be emitted without the presence of particle precipitation. One such example is airglow, which can produce the same 557.7 nm and 630.0 nm emission lines of atomic oxygen as typical aurora, but in this case due to dissociative electron recombination (e.g. Peverall et al., 2000). Interaction between aurora and the dynamics of the neutral atmosphere is a complex subject, with features such as the recently discovered *dunes* potentially being caused by atmospheric
wave modulation on diffuse aurora (Palmroth et al., 2019). Thus, not all emissions similar to aurora are caused by particle precipitation, with Strong Thermal Emission Velocity Enhancement (STEVE) being already a well-known example of aurora-like skyglow, which is likely caused by local acceleration processes instead of precipitation (Gallardo-Lacourt et al., 2018). It is sometimes accompanied by green rays known as picket fence below the purple arc of STEVE (MacDonald et al., 2018). This picket fence is ostensibly related to particle precipitation (Nishimura et al., 2019; Gillies et al., 2019), although some studies
have questioned this connection based on spectral analysis (Mende and Turner, 2019; Mende et al., 2019). STEVE itself has been associated with subauroral ion drifts and local electron heating (MacDonald et al., 2018).

  In this study we suggest Fragmented Aurora-like Emissions (FAEs) as another phenomenon in the same category of aurora-like phenomena, for which particle precipitation is unlikely to be the direct cause. The small fragments of excited plasma discussed in the present study seem to differ from other auroral structures in various ways. They exhibit small horizontal scales
of only a few kilometres, short lifetimes of generally less than a minute and a lack of field-aligned emission extent. Generally, the FAEs occur close to auroral features. This is especially true for FAEs of the second type, occurring in wave-like structures, which were observed with an offset to auroral arcs on the same scale as the FAE size. The next section of the present study aims to provide an overview of the observations and instrumentation used to gather data, followed by a more in-depth description of FAE characteristics. Finally we suggest some potential generation mechanisms and relations to other recently discovered
aurora-like phenomena and summarise our conclusions.

## 2 Instrumentation and observations

All of the analysed FAEs were observed on all-sky camera (ASC) images captured at the Kjell Henriksen Observatory (KHO), which is located on the Breinosa mountain east of Longyearbyen, Svalbard at ∼78.15° N, 16.04° E. The first observation was made on 2015-11-07 between 20:15:58 UT and 20:17:27 UT, with 4 identified FAE candidates over 4 images (further referred
to as "event 1"). FAEs were next seen again on 2015-12-07 between 18:18:14 UT and 18:27:36 UT (20 images, "event 2"), this time a total of 39 candidates. The final observation that is analysed in the present study was made over a much longer period on 2017-12-18 between 07:36:35 UT and 08:26:48 UT, consisting of 79 images ("event 3") in which 262 candidates were marked. Figure 1 shows all these marked candidates for event 3 overlaid on the first image of the series taken at 07:36:35 UT. This is done to visualise the distribution across the sky and general characteristics of the marked candidates, almost all occurred at a
later time during event 3. All FAE events were accompanied by aurora. It should be noted that FAEs have also been sighted at the KHO on at least three other dates, which were more recent and thus not included in the present study.

Due to the availability of varied instrumentation on Svalbard, an effort was made to incorporate many different data sources to obtain FAE characteristics. These include the Sony $\alpha$7S all-sky camera (ASC) and meridian-scanning photometer (MSP) at the KHO, as well as data from the European Incoherent Scatter Scientific Association (EISCAT) Svalbard Radar (ESR) (Wannberg et al., 1997) and high-framerate optical observations with the Auroral Structure and Kinetics (ASK) instrument (Dahlgren et al., 2008) located at the ESR. The ASC images used in the present study have a size of $2832 \times 2832$ pixels. The images were taken using an exposure time of 4 s and an ISO of 16000 at a cadence of 11 to 12 s, with a mean interval length of 11.8 s. This variance is due to variations of the read-out time to the attached computer. The readout delay between the camera and software is responsible for the slower cadence, compared to the camera exposure time. A simple astrometry calibration was used to find the centre of the ASC images and estimate the pixel size, resulting in a scale of 16.59 pixel/degree close to centre. This is further used to determine the offset of FAEs from zenith, which can then be used to calculate the pixel sizes in km for varying elevation angles, using an equidistant projection and an assumed FAE altitude of 110 km. This assumption was based on FAE signatures in the ESR data.

Spectral information is provided by the MSP, which is scanning the auroral emissions at 427.8 nm ($N_2^+$), 557.7 nm and 630.0 nm (both atomic oxygen) with a $1°$ field of view (FOV) from north to south along the local geomagnetic meridian ($31°$ west of geodetic north) using a rotating mirror. Measurements have a time resolution of 8 s (16 s for events 1 and 2), consisting of 4 s (8 s) for a full $360°$ scan plus another 4 s (8 s) for a background scan. Thus, scanning across the sky takes 2 s (4 s). The background measurement is achieved by tilting a narrow band-pass ($\sim$0.5 nm) interference filter for each channel (Chen et al., 2015).

High temporal resolution optical observations from ASK are used to further study the movement and emission properties of the FAEs. ASK consists of three channels with individual band-pass filters for selected auroral wavelengths and lenses to adjust FOV (Ashrafi, 2007). This allows for simultaneous observations of different auroral emissions in a narrow FOV, which can be used to study the energy and flux of the precipitating electrons that produce the aurora (Lanchester et al., 2009). The temporal resolution is 20–32 Hz, and for resolutions above 5 Hz, the available 512 pixels for each camera are binned into a $256 \times 256$ pixel image (Goodbody, 2014). ASK is pointing towards the magnetic zenith and shares part of its observation region with the ESR and the MSP, which led to a finding of a FAE signature in the ESR data after observing its passing across the FOV of ASK. The ASK FOV is $6.2°$ and in this study we use observations of $N_2$ (673.0 nm, first positive band system) and atomic oxygen (777.4 nm) emissions.

Solar wind data from the Advanced Composition Explorer (ACE) and Deep Space Climate Observatory (DSCOVR) satellites at the L1 Lagrangian point can provide insight into the background conditions during the observed events. For the periods preceding the two larger events (2 and 3) the ACE and DSCOVR data show average speeds of 620–640 km/s, which is above the threshold value for high-speed streams (Cranmer, 2002). The $B_z$ component of the Interplanetary Magnetic Field (IMF) is negative and IMF $B_y$ is positive for the relevant periods preceding the FAE occurrences. This indicates an efficient energy transfer into the magnetosphere-ionosphere system. The ACE data for event 1 show average solar wind speeds of $\sim 540$ km/s, negative IMF $B_z$, both of which resemble the other two events to some degree, but negative IMF $B_y$. The $K_p$ indices for the time periods of events 1–3 are 3+, 4- and 4+, indicating moderate geomagnetic activity. Visually inspected convection maps

from SuperDARN radars suggest an ionospheric plasma flow primarily in the northwest or southwest direction. For all our event times Svalbard was located in the evening cell of the convection and close to the flow reversal.

## 2.1 Methods

The FAE candidates appearing on the ASC images were visually identified and manually marked, using the freehand selection tool of the Fiji distribution of the freely available ImageJ software (Rueden et al., 2017; Schindelin et al., 2012). After inspecting the entire image set, the criteria to mark the candidates were identified as outline clarity and strength of the emission intensity enhancement, size, apparent vertical extent and movement across successive pictures. Generally, FAE candidates are clearly offset from the adjacent aurora as emission intensity enhancements confined in a small region, with little to no apparent vertical 95 extent visible in the ASC images. Their limited lifetime results in each individual candidate typically only being visible in 1–4 successive images, with longer lasting candidates showing discernible movement between images. Their short-lived nature often makes identification of newly appearing FAEs relatively obvious when comparing two successive images. Due to the mean cadence of 11.8 s, it is not easy to track FAEs between each image. The term "candidate" in this context refers to a suspected FAE on each individual image, with some of the more stable candidates almost certainly being the same FAE on 100 successive images. While visual identification will certainly introduce some human-observer bias, it is nonetheless a standard approach in auroral studies, since there is no robust automatic identification tool available. It is possible that only the most intense features were identified, but given the large amount of FAE candidates, they should be sufficient to derive the main characteristics of FAEs.

This identification process resulted in a compiled database with all candidates containing their outlines, pixel coordinates 105 and sizes. A total of 305 candidates were marked for further analysis and categorised into 4 confidence groups, depending on their intensity, size and outline characteristics. Group 1 is composed of the most well-defined candidates with clear borders and strong intensity enhancements, whereas candidates in groups 2-4 are of decreasingly lower quality, meaning they are more likely to contain features that are for example part of an auroral arc. The 21 FAEs of the highest quality form group 1, whereas group 2 contains 55 candidates. These 76 candidates are considered as the core set of observations. Group 3 110 contains 78 candidates and group 4 encompasses 151 candidates. FAEs in groups 3 and 4 are analysed in the same manner, but only contribute to the final conclusions if they agree with the core set findings, which would indicate that these are indeed observations of the same phenomenon.

## 3 FAE characteristics

FAEs can be categorised into two distinct categories, the first being individually occurring FAEs. These occur seemingly 115 randomly across the sky, sometimes with a significant offset to the closest auroral arc. The second type are periodic structures with regular spacing between FAEs, which appear close to and generally northwards of auroral arcs. The category 2 FAE group shown in Figure 3 is a typical example.

### 3.1 Distribution, sizes and movement

For the three observed events, most FAEs (73.1%) occurred west of zenith. This is the case for both high- and low-quality candidates, with the dashed kernel density estimation (KDE) in Figure 1 for FAEs of groups 1 and 2 agreeing with the overall distribution KDE. Due to the observational bias caused by the vast majority (262) of FAEs occurring during event 3, this asymmetry in FAE location on the sky might simply be explained by the underlying space weather and ionospheric convection conditions being biased towards westward convection during this period. The low number of FAEs close to zenith (see Figure 1) is possibly explained by observational bias, since FAEs near zenith are harder to identify. Their lack of field-aligned emission extent is not visible when viewed from directly underneath. In addition, most FAEs occurred close to auroral arcs, which rarely appeared close to zenith during the analysed events. The location of category 1 FAEs appears to be fairly random and not necessarily close to auroral arcs, whereas category 2 FAE groups generally appear within the vicinity northwards of an arc, typically with an offset on the scale of the fragment size, corresponding to a few kilometres. Visual inspection of all events

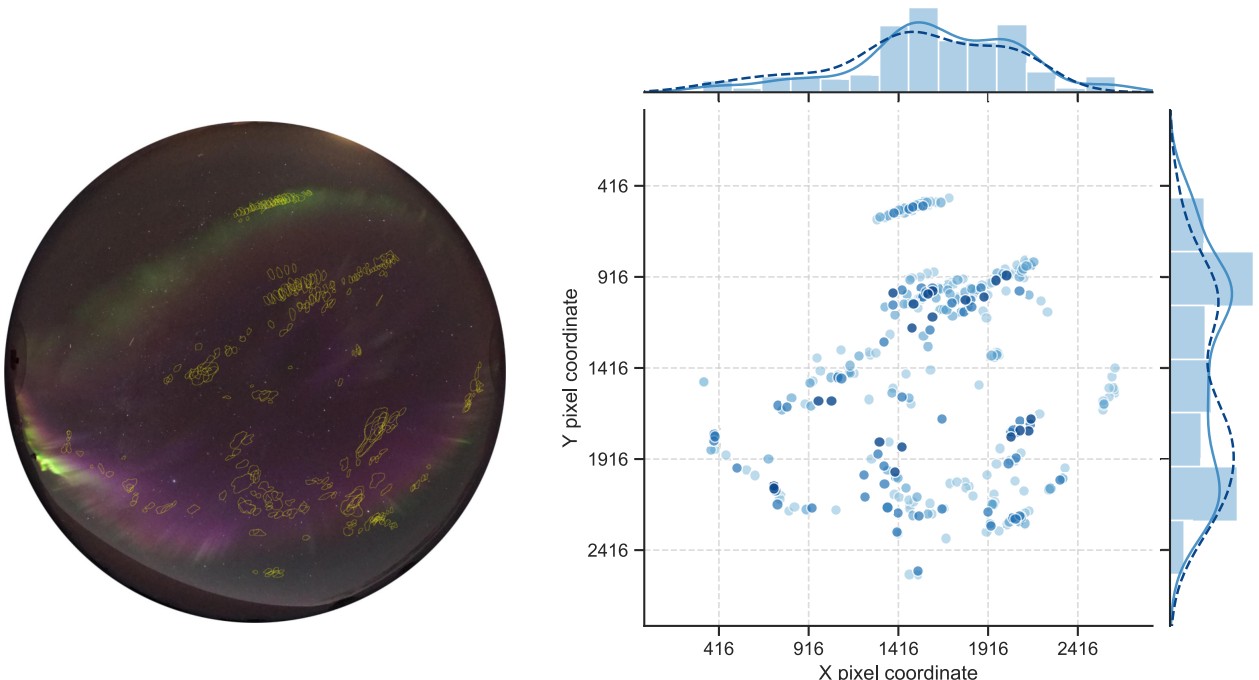

**Figure 1.** Left: All 262 identified FAE candidates for event 3 on 2017-12-18, overlaid on the first image of the series taken at 07:36:35 UT. The FAE candidates occurred over a time period of ∼50 minutes. Geomagnetic east corresponds to the left side of the image, geomagnetic north to the top. Right: All 305 FAE locations in horizontal and vertical pixel coordinates with histogram distribution and kernel density estimation (KDE). FAEs are shaded according to confidence groups, with darker shades being FAEs of higher quality. The dashed KDE line is only calculated for FAEs of groups 1 and 2. Credit: ASC image provided by the KHO.

shows that FAEs appear mostly elliptical, thus fitting an ellipse to follow the marked outline of each FAE provides a more

robust estimate of its size. As shown in Figure 2, the fitted ellipses of most FAEs have a major axis of 20 km or less, with a few larger outliers that might simply be diffuse auroral patches, especially on the larger end of the marked size range. The average major axis length is ~6–8 km, with an average minor axis of ~3–4 km. Their aspect ratio (AR = [Major axis]/[Minor axis]) has a mean value of 2.04. Most FAEs seem to have fairly regular, rounded shapes with few indents, with a mean circularity value of $c = 0.705$ ($c = 1$ being perfectly circular), which is determined using the formula $c = 4\pi \cdot [\text{Area}]/[\text{Perimeter}]^2$. This

determination is of course affected by their size, with deviations from rounded shapes being harder to identify in smaller FAEs, with an added general operator bias to outline regular shapes compared to complex indents. It should be noted that due to the 4 s integration time of the ASC, any fast-moving object will appear somewhat elliptical. Nevertheless, this is not true for the high-framerate data from ASK, which also show FAEs to be elliptical. The described trends are observable in both high- and low-quality candidates, as KDEs for high-quality FAEs are in good agreement with the entire data set in Figure 2. This suggests that most of the marked candidates of groups 3 and 4 are indeed FAEs. Category 2 FAEs can be seen moving along

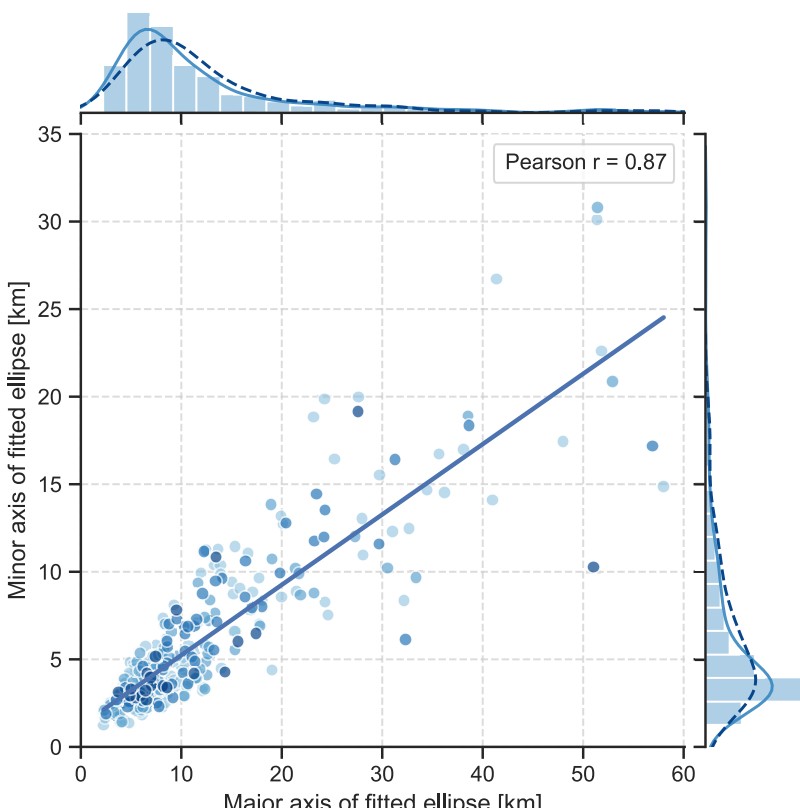

**Figure 2.** Length of major and minor axes (in km) of fitted ellipses for each FAE, assuming an altitude of 110 km. FAEs are shaded according to confidence groups, with darker shades being FAEs of higher quality. A histogram of the variables is plotted on the outer axes, together with a KDE. The dashed KDE line is only calculated for FAEs of groups 1 and 2. The legend shows the calculated statistical Pearson correlation coefficient for a linear regression, with a p-value $\ll 0.01$, rejecting the null hypothesis.

the auroral arc in Figure 3. The distance between these FAEs does not vary significantly as they move eastward over a period of 35 seconds. A spatial intensity variation is visible in the grouped structure, where FAEs appear dim towards the edges of the group and become more intense the closer they move towards the centre. Some of the variation in intensity seems to be caused by fragments appearing and disappearing at the ends of the group. Using an average distance of 45 pixels between the FAEs and their approximate elevation angle of $\sim 65°$, we can roughly estimate the spacing between FAEs for this group to be around $\sim 6$ km. Visual inspection of the ASC images shows a general westward movement of the FAEs for the observed events, which might originate from the underlying convection pattern. No obvious eastward motion was observed. A few FAEs were observed in the ASK high-framerate images (see Figure 5), with some remaining stable for multiple seconds while they drift, whereas others appeared and vanished within a second. The ASK FOV corresponds to $10 \times 10$ km$^2$ at an altitude of 100 km, which FAEs passed within $\sim 10$–14 s. This results in an estimated drift speed on the order of $\sim 1$ km/s.

## 3.2 Observed emissions

For FAE positioned along the MSP scanning line, the MSP data were checked to search for corresponding signatures. Three FAE signatures were found, of which one is presented in Figure 4. Distinct FAE emissions were observed at the 557.7 nm (green MSP channel) line of atomic oxygen, but not at the 630.0 nm (red channel) line of atomic oxygen, nor at the 427.8 nm (blue channel) emission of $N_2^+$. Due to the long lifetime of the 630.0 nm emission state ($\sim 110$ s) and the short-lived and fast-moving nature of FAEs, the respective MSP red channel measurements are unlikely to show any distinct FAE signatures, with any potential emissions "smeared" over the temporal axis. Figure 4 shows a clear peak at the FAE elevation of $\sim 100°$ in the 557.7 nm measurements while it passed the MSP scan line (marked by vertical lines), with a clear drop-off as the FAE moved out of the scan and faded. No distinct signature can be seen at this elevation in the 427.8 nm measurements. A broad general increase is visible over a large area in the 630.0 nm emissions, likely by the background aurora at higher altitudes, as this emission was elevated before and after the FAE occurrence. Also, at the suggested FAE altitude of $\sim 110$ km, the atomic oxygen state which emits at 630.0 nm is heavily collisionally quenched and thus any FAE emissions at this wavelength at low altitudes are expected to be extremely weak. It should nonetheless be noted that the broad increase may potentially hide a FAE signature in the 630.0 nm data. The other MSP passings show comparable results.

One FAE was observed passing through the ASK FOV during event 2 on 2015-12-07 (for the corresponding video file see Whiter, 2020). The ASK instrument provides temporal and spatial high-resolution observations. $N_2$ emission signatures at 673.0 nm (first positive band system) in the ASK channel 1 data can be seen in the left and middle panel in the bottom row of Figure 5. At the same time, no emission is visible in the right panel in the bottom row, which shows the ASK 3 channel measuring at 777.4 nm (atomic oxygen). The ratio between 777.4/673.0 nm emissions is commonly used to determine the energy of precipitiating particles, and typically the lack of 777.4 nm emissions resulting in very small ratios would mean high energy precipitation (e.g., Lanchester et al., 2009; Dahlgren et al., 2016). But even with very high energies, there should be some 777.4 nm as well as 427.8 nm emissions. The apparent lack of these emissions suggests a different generation mechanism than precipitation. As the FAEs show emissions at 557.7 nm and 673.0 nm, but seemingly not at 427.8 nm or 777.4 nm, looking at the excitation thresholds of these emissions can give a clue towards the upper energy limits of the generation mechanism.

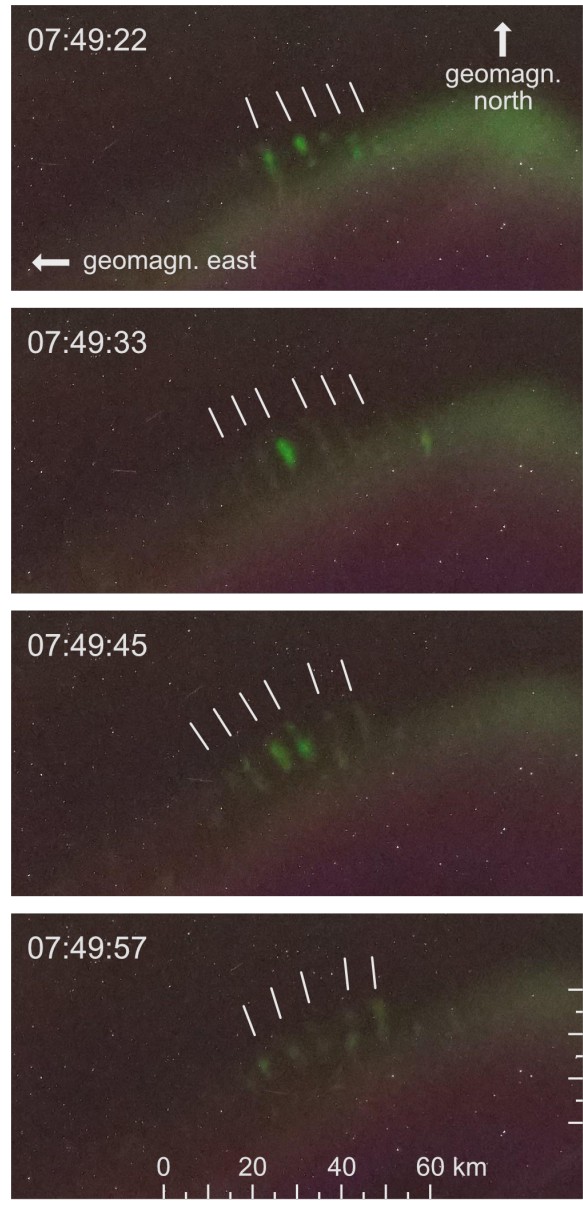

**Figure 3.** Movement of a category 2 FAE group northwards of the main auroral arc (northwest of zenith) over four successive images taken on 2017-12-18 around 07:49:40 UT. The images are cropped to $1000 \times 500$ pixels to make the FAEs easily identifiable. White lines indicate the apparent alignment of the FAEs and were used to determine approximate distances between them. A scale in kilometre is added for reference, using a pixel to km ratio of 0.129 (at $65°$ elevation angle). Credit: ASC images provided by the KHO.

Excitation thresholds for the 427.8 nm and 777.4 nm emissions lie above 10 eV (e.g., Lanchester et al., 2009; Holma et al., 2006), with the lowest possible excitation energy being $\sim$11 eV for direct excitation of atomic oxygen at 777.4 nm. For the

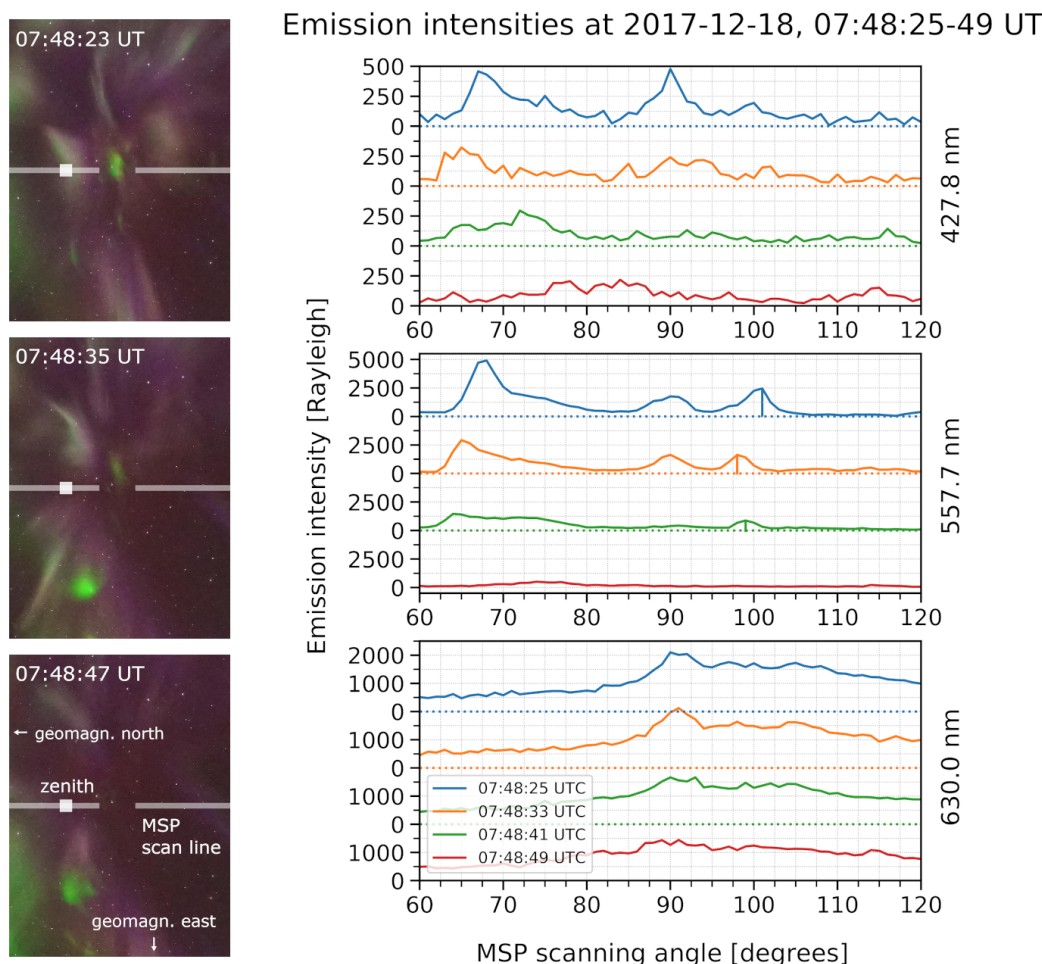

**Figure 4.** Comparison of consecutive cropped ASC images and MSP line scans for a FAE moving through the MSP scan line on 2017-12-18 between 07:48:23–47 UT. The FAE signatures are marked with vertical lines in the green channel (557.7 nm). The MSP scan line (1° width) is drawn on the ASC images in grey. A grey square marks the geographic zenith in the centre of the ASC images. Credit: ASC images provided by the KHO.

observed 557.7 nm and 673.0 nm emissions the excitation energies are 2 and ~8 eV, respectively (e.g., Holma et al., 2006; Ashrafi et al., 2009). Combined, this suggests an upper limit for the energy of the generation mechanism between ~8–11 eV.

### 3.3 Plasma characteristics measured with the ESR

To further understand the underlying plasma properties of FAEs, an attempt was made to find signatures within incoherent scatter data of the ESR. The auroral arc visible south of the FAE in Figure 5 extended across the entire FOV of ASK (partially shared with the ESR) shortly before the FAE occurrence at 18:23 UT, and is visible in Figure 6 as a general increase in electron

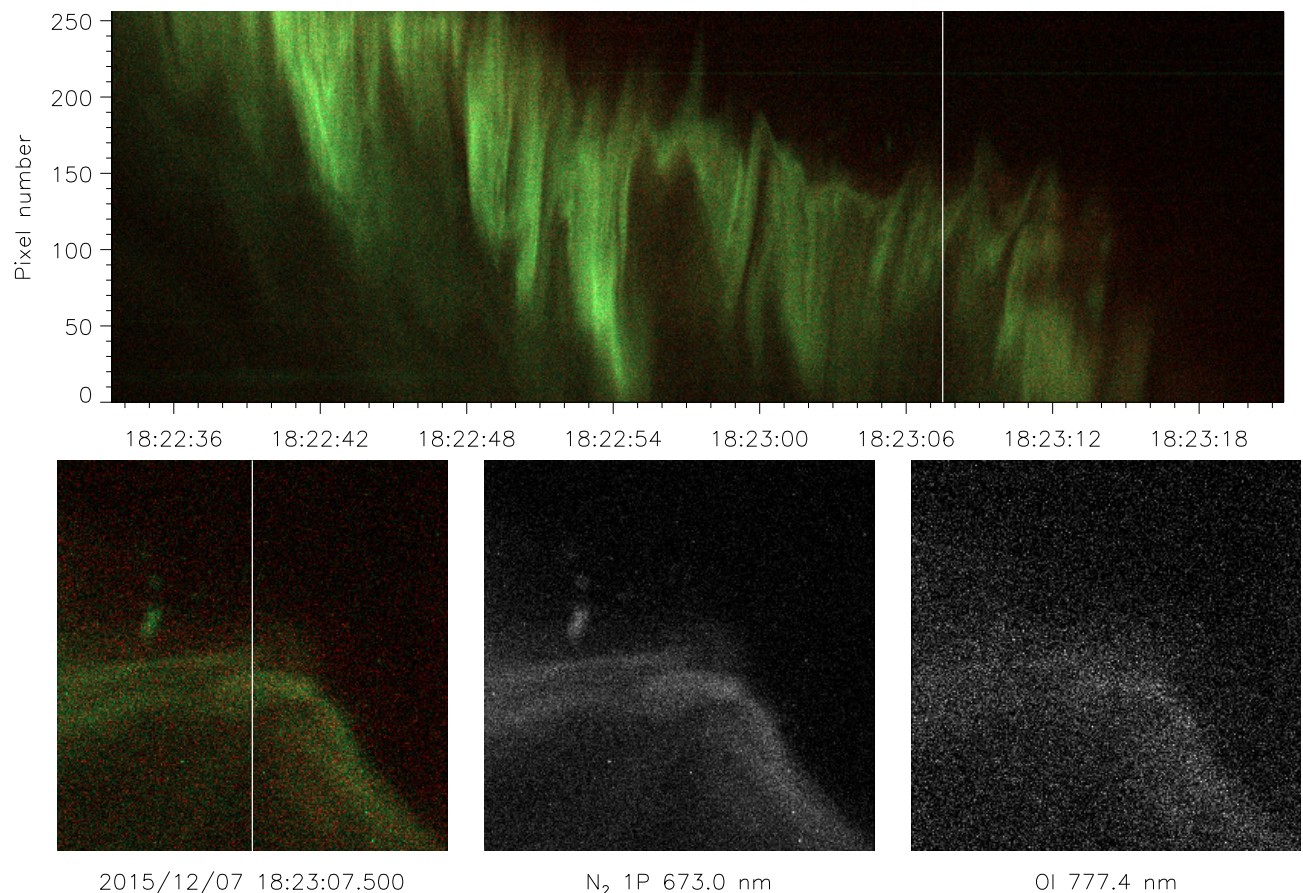

**Figure 5.** ASK keogram for the event of 2015-12-07 around 18:23:07 UT in the upper panel. ASK1 measuring the 673.0 nm emission of the first positive band system of $N_2$ is visible in the lower middle panel, the lower right panel shows the ASK3 measurement of 777.4 nm emissions of atomic oxygen, and the lower left panel shows ASK1 in the green/blue channel and ASK3 in the red channel.

density across the entire altitude range. The density decreases across most altitudes as the arc moves out of the FOV towards 18:23 UT. It remains high at 113 km at the time of the FAE occurrence. No associated increase in electron temperatures is visible in Figure 6 for the period and altitudes of the arc signature in the electron density panel.

The FAE visible in Figure 6 shows as a local increase in electron temperature to ∼2300 K at 113 km around 18:23 UT. This increase seems to be confined to a narrow altitude range, which is further established by the time series at four successive altitudes shown in Figure 7. The increase at the time of the FAE passing is limited to altitudes below 119 km and strongest at 113 km. For the period directly after the FAE occurrence, multiple increases in electron temperature are visible at low altitudes, which indicates an unstable lower ionosphere. Simultaneous increases in ion temperatures are visible at higher altitudes, with

significant increases around 190 km, up to ∼4500 K.

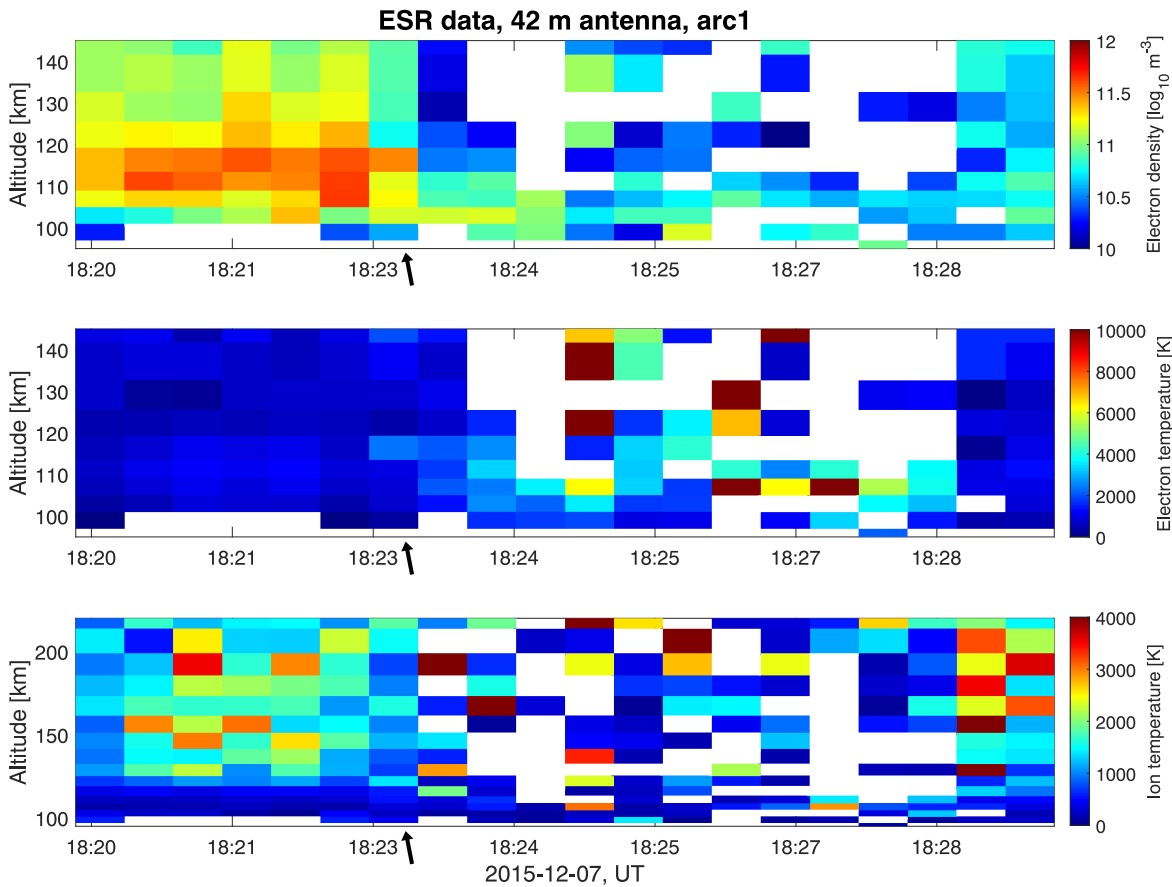

**Figure 6.** Incoherent scatter data from the ESR (analysed with GUISDAP) for 18:20–18:30 UT on 2015-12-07, with electron densities in the upper, electron temperatures in the centre and ion temperatures in the lower panel. Data points with errors > 50 % of the values were removed. Further limiting to > 30 % would only remove a few extra data points. Errors for the relevant time periods up to the FAE passing are < 20 % of the values. The arrows mark the time of the FAE passing.

The background conditions during these analysed events might be able to further provide some insight into the underlying generation mechanism. For the entire duration of event 3, significant intermittent increases in electron temperatures were observed at altitudes in the E-region, as well as elevated ion temperatures (mostly) in the F-region. This indicates a connection between FAEs and elevated electron temperatures at low altitudes, which we will discuss below.

## 4 Discussion

Fragmented aurora-like emissions have been analysed and classified in the present study, with results suggesting that they are a new type of aurora-like feature. Comparing FAEs with ostensibly similar auroral phenomena shows some key differences.

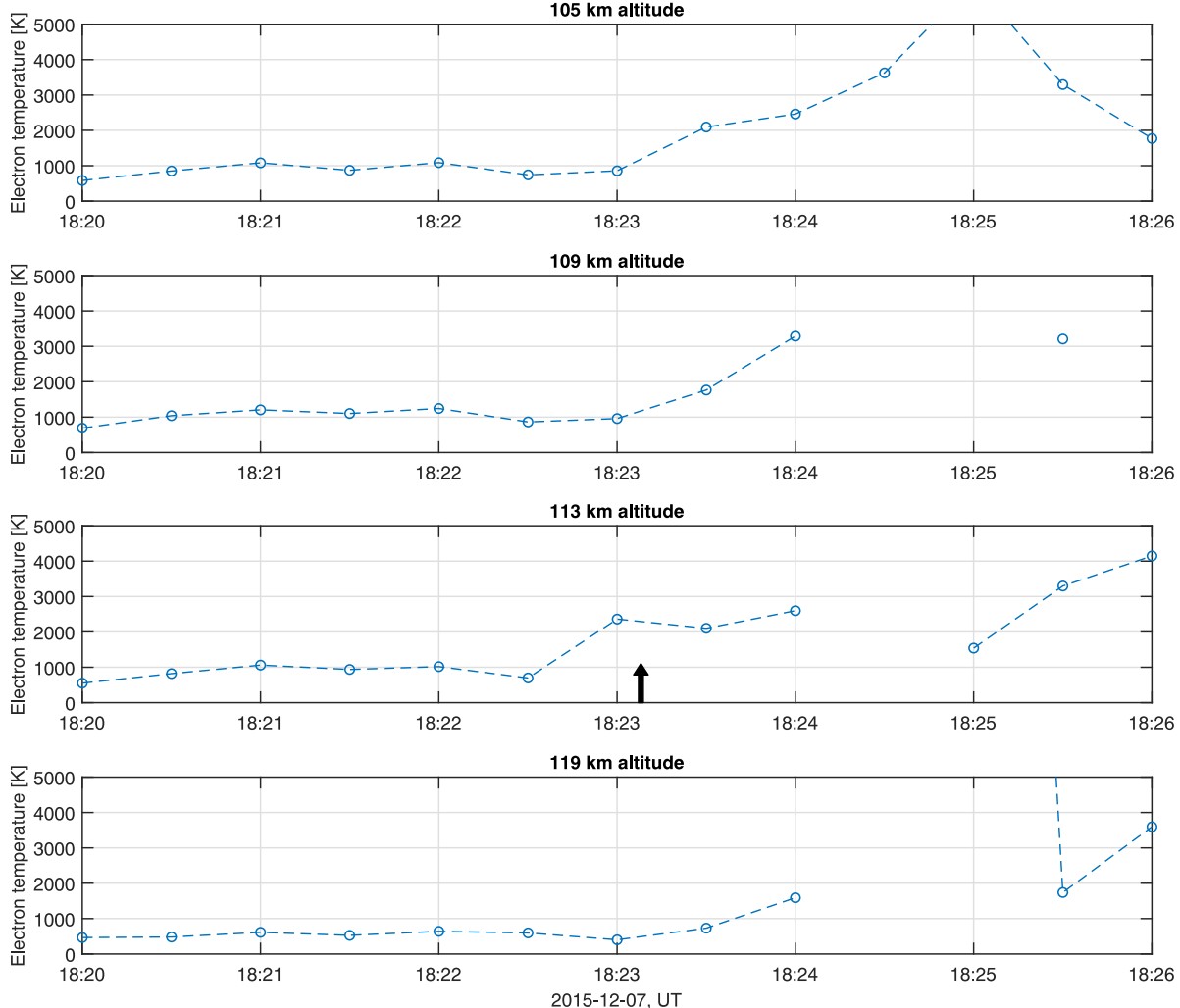

**Figure 7.** Time series of electron temperatures at four successive altitudes between 105–119 km from incoherent scatter data from the ESR (analysed with GUISDAP) for 18:20–18:26 UT on 2015-12-07. Data points with errors > 50 % of the values were removed. The arrow marks the time of the FAE passing and denotes the distinct increase in electron temperature specifically at 113 km.

For example, the term enhanced aurora (EA; see Hallinan et al., 1985) describes an enhanced emission in a thin layer, typically along a rayed auroral structure. Albeit also designating a localized emission intensity enhancement occurring alongside aurora, EA differs in various characteristics. EA occurs as layers with limited vertical extent, but longitudinal and latitudinal extents of at least 250 km and 300 km, respectively (Hallinan et al., 1985). FAEs are much smaller, with minor and major axes sizes of < 10 km. While EA manifests as intensity enhancements along the rays of a bigger auroral feature, FAEs were clearly dislocated from the field lines of the adjacent rayed structures. FAEs also lack the blue emission enhancement visible in EA.

Furthermore, EA has been observed as quasi-stable structures lasting for minutes, while most analysed FAEs had lifetimes of
less than a minute. Overall, this suggests that these are two different phenomena.

When comparing FAEs with pulsating patches, two major distinctions between the two phenomena are size and lifetime of the individual features. Pulsating patches occur within diffuse aurora, whereas the analysed FAEs are seen alongside discrete arcs. FAEs are much smaller than pulsating patches, which are also typically very stable, while showing quasi-periodic fluctuations in their emission intensity (e.g., Humberset et al., 2018; Nishimura et al., 2020). In contrast, FAEs are short-lived and do not show any emission intensity fluctuations, apart from appearing and fading away. The available ASK video observations of FAEs show their much higher dynamic motion and smaller size, compared to pulsating patches. Together, these differences lead us to conclude that FAEs are a distinctly different phenomenon.

As the FAEs were found by manual inspection of images, there is some bias in which features were selected and how they were classified. The data set could contain other auroral small-scale forms or diffuse patches, which is the reason for the classification into four confidence groups. As the general properties of candidates between high- and low-confidence groups agree well, we are confident that most selected features are indeed FAEs. Generally, FAEs can be distinguished from other auroral forms by their lack of field-aligned emission extent, as suggested by the off-zenith parts of the ASC images and field-aligned ionisation measured by the ESR, small sizes and short lifetimes. A FAE signature is visible in the ESR data as locally enhanced electron temperatures around 113 km. Determining a definite FAE altitude requires triangulation, which was not possible for the analysed ASC images, or other means of consistently identifying FAE signatures in measurements over an altitude range, such as multiple signatures in EISCAT data.

Semeter et al. (2020) recently described green "streaks" below STEVE, which show various similarities to FAEs. Their triangulation positions the streaks at an altitude of 100–110 km, which is also the region we suggest FAEs to occur within. They propose superthermal electrons resulting from the extreme electric fields during STEVE as a local generation mechanism, similar to our hypothesis. It will be interesting to see if these two phenomena are indeed related on a fundamental level, or just bear superficial resemblance. Gallardo-Lacourt et al. (2018) suggest STEVE as another locally generated skyglow without any associated particle precipitation. The phenomenon is far from well-understood and occurs on much larger scales than FAEs, but indicates that ionospheric processes can indeed cause emission without particle precipitation being present. We propose that FAEs fall within the same category, even though many of their properties such as size and lifetime differ majorly. The underlying processes heating the plasma are unlikely to be the same, but on a fundamental level both emissions seem to be related to thermal ionospheric processes rather than particle precipitation.

The present study aims to present the basic characteristics of FAEs and categorise them based on the three analysed events. Nonetheless, the available data enable us to hypothesise about their underlying generation mechanism. The analysed events show above-average solar wind speeds (except for event 1), negative IMF $B_z$ and positive $B_y$, with a westward convection of FAEs. They are not limited to a certain time sector, with occurrences both between 10:30–11:30 MLT and 21:15–23:15 MLT. The elevated electron temperature at E-region altitudes and simultaneous increases in ion temperatures at higher altitudes can provide some clues towards the origin of FAEs. One possible group of generation mechanisms for the required energy to excite FAEs are Farley-Buneman instabilities, which are streaming instabilities typically occurring at altitudes of 90–120 km (Oppen-

heim et al., 1996). The proposed FAE altitude falls within this region. They become significant when the difference between
electron and ion drift speeds exceeds the ion acoustic speed (Liu et al., 2016), which is generally the case in geomagnetically
disturbed conditions, typically also resulting in aurora. This would explain why FAEs are observed alongside aurora. Particu-
larly at high latitudes, these instabilities can result in significant local electron heating. This is consistent with the low-altitude
elevated electron temperatures observed during the FAE events, for which Farley-Buneman instabilities are the most likely
explanation.

The observed large ion temperatures in the F-region around 190 km height are caused by Joule heating from strong electric
fields, or ion-neutral friction. The measurements are used to estimate the electric field strength below assuming that $E_\parallel = 0$,
i.e. the magnetic field-lines are equipotentials. We neglect the effect of the slightly different magnetic field strengths between
190 km height and the lower E-region, and also any differences of the neutral wind between these altitudes. The ion energy
balance, neglecting also thermal energy transfer to/from electrons (whose temperatures are generally not enhanced above the
E-region, especially preceding the FAE occurrence at 18:23 UT) is (Alcayde et al., 1983, Equation (4)):

$$Q_{in} = \nu_{in} N_n N_e \left( \frac{3}{2} \, k_B \left( T_n - T_i \right) + \frac{1}{2} \, m_i \left( \boldsymbol{V}_i - \boldsymbol{V}_n \right)^2 \right) \tag{1}$$

Here $T_i$ and $T_n$ are the ion and neutral temperatures, $\boldsymbol{V}_i$ and $\boldsymbol{V}_n$ the ion and neutral drifts, respectively. $m_i$ is the mean ion
mass, $k_B$ the Boltzmann constant, $\nu_{in}$ the ion-neutral collision frequency, and $N_n$ and $N_e$ the neutral and electron densities. In
the steady state $Q_{in} = 0$, and for the F-region we insert $\left( \boldsymbol{V}_i - \boldsymbol{V}_n \right) = \boldsymbol{E}_\perp \times \boldsymbol{B} / B^2$ with $\boldsymbol{E}_\perp$ the electric field in the frame of
the neutral gas and $\boldsymbol{B}$ the geomagnetic field. We are only interested in the magnitude of $\boldsymbol{E}_\perp$, which can be estimated as

$$E_\perp / B = \sqrt{3 \, k_B \left( T_i - T_n \right) / m_i} \tag{2}$$

Filtering out elevated ion temperatures above 1500 K, we use the ESR data to estimate a mean background ion temperature in
the quiet state of $\sim$950 K for the altitude range of 150–300 km, which should then approximately correspond to the neutral
temperature. For $m_i$ we use conservatively 30 amu, corresponding to a mixture between $N_2^+$ and $O_2^+$, neglecting a contri-
bution by $O^+$. The motivation is that high $T_i$ and large drift difference $|\boldsymbol{V}_i - \boldsymbol{V}_n|$ probably enhance the relative molecular
concentration compared to model values as the International Reference Ionosphere (IRI) would give it. Using average elevated
$T_i \approx 3300$ K for the altitude range of 150–300 km from the ESR measurement, the estimated lower limit is $E_\perp \approx 70$ mV m$^{-1}$.
This value is far above the threshold for Farley-Buneman instabilities, which is typically around 30 mV m$^{-1}$ (Williams et al.,
1992). If molecular ions were assumed to be dominant, it would only further increase the lower limit. It should be noted that
this is an approximation and the filtering for average values is based on somewhat arbitrary choices, but the derived $E_\perp$ is not
all that dependent on the inserted $T_i$ and $T_n$ and would exceed the typical limit for Farley-Buneman instabilities by a significant
margin regardless of the exact filtering values. The threshold may be already exceeded in the arc, before 18:23 UT, but $T_e$ was
perhaps not high enough to excite optical emissions. Buchert et al. (2008) showed an example with the ESR where $T_e$ reaches
temperatures above 3000 K at 100–109 km, which is enough to produce 630.0 nm optical emissions according to Gustavsson
et al. (2001). An open question is whether these instabilities can produce large enough $T_e$ increases to excite all observed FAE
emissions. Buchert et al. (2008) showed that $T_i$ increased already above $\sim$125 km, up to about 4000 km. These temperature

enhancements are stronger than those observed at auroral oval latitudes over mainland Norway by Williams et al. (1992). This could be because at the edge of the auroral oval over Svalbard E-fields may be larger than in the auroral zone, or because the ESR is more sensitive than the EISCAT mainland radar was in 1992. If E-fields (and associated $T_e$ enhancements) are typically larger at Svalbard, this might perhaps explain why FAEs have not been noticed earlier in the auroral zone or the Scandinavian mainland. Another possible contributing factor could be that auroral all-sky cameras used for scientific purposes are often more limited in pixel resolution compared to the Sony $\alpha$7S used in the present study, which could reduce the likelihood of unexpectedly identifying small-scale and short-lived features like FAEs.

Whereas specific characteristics for the individually occurring FAEs are hard to identify, category 2 FAE groups with regular spacing clearly suggest a link to wave activity. We tentatively suggest that waves modulate the electric field strength and correspondingly the intensity of Farley-Buneman induced plasma turbulence and electron heating near the arcs to produce the observed category 2 FAE groups. As these groups show regular and fairly stable distances between the individual FAEs, some kind of monochromatic wave seems to be responsible. Suzuki et al. (2009) describe the modulation of airglow by gravity waves, which is similar to the modulation of category 2 FAE groups, albeit at larger scales. The short distances between FAEs suggests waves with small wavelengths. The estimated FAE drift speed of ∼1 km/s is much faster than the average ionospheric convection speed of a few hundred m/s. If category 2 FAEs are indeed modulated by waves, they could propagate with their phase velocity and thus exceed typical convection speeds. Alternatively the E-field modulation could originate from the magnetosphere. A candidate mechanism is that the shear between the strong flow in the high E-field adjacent to the arc and the slower flow in the arc itself leads to a Kelvin-Helmholz instability, whose phase speed would be between the slow and fast flows (see, e.g., Keskinen et al. (1988)). For $E_\perp \approx 70$ mV m$^{-1}$, corresponding to 1400 m/s, the phase speed of Kelvin-Helmholtz waves would be several hundred m/s, which is roughly the observed value. It is, however, unclear why the auroral arc shows no signature of the modulation, and what determines the wavelength of the quasi-periodic FAEs of ∼6 km.

To determine a link between FAEs and other aurora-like features like STEVE or the green "streaks", and to further analyse FAE characteristics, more events will need to be studied, ideally from multiple locations and with ionospheric plasma measurements. The limited sample size, not necessarily of FAEs, but rather observation nights and ESR data for the present study, limits the conclusions that can be drawn for the underlying generation mechanism. Until these conditions are determined, FAE occurrences will be seemingly random, further complicating a targeted follow-up study.

## 5  Conclusions

The focus of the present study is to characterise a new type of aurora-like phenomenon, which we name Fragmented Aurora-like Emissions (FAEs). In summary, the observed FAEs can be grouped into two categories: individually occurring FAEs and groups close to auroral arcs with wave-like structure. All FAEs show a lack of field-aligned extent and seem to generally occur in the shape of an elongated ellipse. The majority of the observed FAEs have a major axis smaller than 20 km (assuming an altitude of ∼110 km), with a mean aspect ratio of ∼2. Photometer data show distinctly enhanced intensities at the 557.7 nm emission of atomic oxygen for FAEs passing the FOV, but no clear FAE signatures at the 427.8 nm and 630.0 nm wavelengths,

of which the latter is not surprising, as it would be heavily collisionally quenched at the proposed altitude. A FAE signature is also clearly visible in the ASK1 673.0 nm emission channel of the first positive band system of $N_2$, but not at the 777.4 nm emission of atomic oxygen measured by ASK3, which together sets a range of states with different energies that are excitable by the generation mechanism. The apparent lack of 427.8 nm and 777.4 nm emissions indicates an upper energy limit between $\sim$8–11 eV which the generation mechanism can produce. The ESR data suggest that FAEs are associated with significantly elevated electron temperatures around 110–120 km, for which Farley-Buneman instabilities are the only known cause at these low altitudes. Simultaneously, increased ion temperatures are visible at altitudes in the F-region, which enables us to estimate the strength of the E-field. The derived estimate of $E_\perp \approx 70$ mV m$^{-1}$ exceeds the typical Farley-Buneman threshold of 30 mV m$^{-1}$. Category 2 FAE groups show a fairly regular and stable spacing and appear to be modulated by some kind of wave.

Open questions are the exact nature of the generation mechanism, whether FAEs of categories 1 and 2 are caused by the same mechanism, if category 2 FAEs are indeed modulated by wave activity and if so by what kind of wave, whether they are exclusively a high-latitude phenomenon and what threshold values of ionospheric parameters are necessary for FAE occurrences.

*Data availability.* ACE data are available on the website of the ACE Science Center (http://www.srl.caltech.edu/ACE/ASC/). DSCOVR data are available from the NOAA Space Weather Predicition Center (http://doi.org/10.7289/V51Z42F7). SuperDARN data are available on the website of Virginia Tech (http://vt.superdarn.org/). ASC and MSP data are available from the KHO website (kho.unis.no). ASK data are availble from the ASK teams at KTH Stockholm, Sweden and the University of Southampton, UK. EISCAT data can be downloaded from the MADRIGAL database: http://portal.eiscat.se/madrigal/

*Video supplement.* Whiter (2020) provides access to the ASK video on which Figure 5 is based.

*Author contributions.* JD analysed the data set and wrote the present study. NP contributed towards the entire writing and analysis process. DW suggested the FAE name and contributed towards the writing and data analysis process, especially regarding the ASK data. PGE originally discovered the FAEs in ASC images and contributed towards the writing and data analysis process, especially regarding the ASC and MSP data. LB contributed towards the ESR data analysis and respective section. SCB suggested Farley-Buneman instabilities as a potential generation mechanism and contributed the respective discussion section.

*Competing interests.* NP and DW are editors for the special issue this paper has been submitted to.

*Disclaimer.* This study is based on J. Dreyer's master's thesis (Dreyer, 2019), which in parts contains some additional information that might be of interest.

*Acknowledgements.* FAEs were independently identified in ASK data by Hanna Sundberg for an event in 2013. JD is thankful for being supported by the Swedish National Space Agency under grant Dnr 143/18. The work by NP & LB is supported by the Norwegian Research
Council (NRC) under CoE contract 223252. DW is supported by the Natural Environment Research Council (NERC, UK) under grant NE/S015167/1. ASK has been supported by NERC of the UK under grants NE/H024433/1, NE/N004051/1 and NE/S015167/1. The authors thank the KHO team and PI Dag Lorentzen for maintenance and calibration of the Sony camera and MSP. SuperDARN is a collection of radars funded by the national scientific funding agencies of Australia, Canada, China, France, Japan, Norway, South Africa, United Kingdom, and United States of America. EISCAT is an international association supported by research organisations in China (CRIRP), Finland (SA),
Japan (NIPR and ISEE), Norway (NFR), Sweden (VR), and the United Kingdom (UKRI).

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
