# Peer review of "Fragmented Aurora-like Emissions (FAEs) as a new type of aurora-like phenomenon"

_Annales Geophysicae, 2020_

## Referee Comment (RC1) · Anonymous Referee #1 · 7 Sep 2020

General Comments

The paper is well-written and organized. It presents interesting observations of short living small scale aurora-like structures of high scientific interest. The presented first summary for characteristic features of the discussed Fragmented Aurora-like Emissions is important for future follow-up studies. Instrumentation, observations and methods are well explained. The paper presents images and spectral data for FAEs strongly supporting the author's hypothesis for a low energy generation mechanism with an upper energy limit between ~8-11 eV which excludes a formation caused by precipitating electrons. The authors clearly state that the exact generation mechanism remains unclear. Their finding that FAEs are associated with elevated electron temperatures points to Farley-Buneman instabilities as a potential energy source and sets an important base for follow-up studies. I have only a few minor comments for the authors to consider a few minor additions prior publication.

Specific comments

Major comments: No major comments.

Minor comments:

- It would be helpful to add a video showing an example for a category 2 FAE. - L.29–30: I recommend to add references for the following papers all presenting strong arguments against the hypothesis that precipitating electrons are responsible for picket fence structures below the purple arc of STEVE (Nishimura et al., 2019). Paper 1: Gillies D. M. et al. (2019). First Observations From the TREx Spectrograph: The Optical Spectrum of STEVE and the Picket Fence Phenomena, Geophysical Research Letters, 46 (13), 7207–7213. Paper 2: Mende S. B. & Turner C. (2019). Color Ratios of Subauroral (STEVE) Arcs, Journal of Geophysical Research: Space Physics, 124 (7), 5945–5955. Paper 3: Mende S. B., et al. (2019). Subauroral Green STEVE Arcs: Evidence for Low-Energy Excitation, Geophysical Research Letters, 46 (24), 14256–14262. - L.47–48: The authors mention that similar structures (FAEs) have been sighted on Svalbard at other days. I recommend to mention on how many days FAEs have been identified. - L. 53–56: [...The images were taken using an exposure time of 4 s and an ISO of 16000 at a cadence of 11 to 12 s, with a mean interval length of 11.8 s. This variance is due to variations of the read-out time to the attached computer, with the camera exposure time set to 10 s...] Contradicting exposure times. What is correct, 4 s or 10 s? Please clarify. - Figure 4: This figure shows a mark for the zenith. Is this the local magnetic zenith? Please clarify. - Figure 6 and 7: [...Data points with errors > 50% of the values were removed...] What are the errors for the shown data points? Are they close to 50% or significantly less? Please clarify.

Technical corrections:

None
* * *

---

## Referee Comment (RC2) · Anonymous Referee #2 · 24 Sep 2020

In general, I disagree with the premise of the paper to create a name for an aurora-like phenomena, because the aurora in general contains so much natural variation and on a continuum of spatial and temporal scales. However, this does not invalidate the careful and thoughtful work that the authors have done in analyzing the available data for several specific auroral events. I do not want this paper rejected right out, but I would like to see less emphasis on trying to establish a new name for an aurora-like phenomenon and more emphasis on the analysis of small-scale auroral features, which show interesting aspects when analyzed in such detail, for example multiple processes for electron acceleration happening in close proximity or even on the same field line at different places. It seems that the features discussed only happen in conjunction with aurora and are thus part of the aurora.

[Figure]

Specific comments:

Figure 1 does not show any clear evidence of anything other than a typical auroral display. Also, the actual features cannot be seen under the yellow areas overlaid on the image.

Figure 3 also does not show anything convincing either. There is no scale to gauge the size of these features extracted from the all-sky camera images. How do these features differ from what has been termed enhanced aurora (see Hallinan, et al., 1985)?

Section 2.1: Inadequate description of how these are identified (just identified by eye). This will generate selection biases and their identification in general is based on some thresholds visible in the images that depend on the sensitivity of the camera and the eye. For example, if a more sensitive camera were used, it is possible that a diffuse background of aurora would become visible and these spots are just localized enhancements of that background. How are they identified, what metrics are used to determine their boundaries, identifying them by eye is not good enough...and can easily introduce errors and biases.

Section 3.1: The paper mentions that the larger patches identified might just be diffuse auroral patches. Is it not possible that all of these FAEs might just belong to the general category of diffuse aurora? Pulsating / diffuse auroral patches have been found to have very limited altitude extent (see Stenbaek-Nielsen and Hallinan, 1979) and is believed to be a fairly common feature among the diffuse auroral structures.

Line 152: Can you provide a reference that discusses the details of the electron energy estimate from that emission line ratio?

Line 180: I would not say that it is 'clear'. The data presented seem to just show normal variations within the diffuse aurora.

Line 184-185: It is not clear how the 'field-aligned emission extent' is measured. Parallax is not used, so is it just the off zenith viewing geometry of most of the all-sky FOV?

If the latter is the case, there will likely be large uncertainty in the altitudes just based on the viewing geometry and what other auroral features could lie along the same line of sight at different altitudes. If it is the EISCAT signatures, then the wording should reflect the altitude of ionization and not auroral emissions.

Line 215-216: Is it not still possible that the O+ density could be up to ∼10 times higher than the N2+ and O2+ densities in this altitude range?

Lines 253-257: This paragraph summarizes the overall uncertainty and limitations of the data used in the current study, which limits the conclusions that can be drawn from them. Thus it is not scientifically sound to define a new feature with such a limited data set, especially without any clear metrics of how they are defined. Multitudes of auroral structures have been observed within both discrete and diffuse aurora for many decades and only very few have been assigned specific names and those come mostly from historical reasons.

——————— References: Hallinan, T. J., H. C. Stenbaek-Nielsen, and C. S. Deehr (1985), Enhanced aurora, J. Geophys. Res., 90, 8461–8475, doi:10.1029/JA090iA09p08461.

Stenbaek-Nielsen, H. C. and T. J. Hallinan (1979), Pulsating Auroras: Evidence for Non-collisional Thermalization of Precipitating Electrons, J. Geophys. Res., 84, 3257-3271.

---

## Author Comment (AC1) · 21 Oct 2020

AR: We thank the referee for providing their helpful feedback! In the following, we respond (Authors' Response, blue) to each of the referee's comments (black) individually.

General Comments

The paper is well-written and organized. It presents interesting observations of short living small scale aurora-like structures of high scientific interest. The presented first summary for characteristic features of the discussed Fragmented Aurora-like Emissions is important for future follow-up studies. Instrumentation, observations and methods are well explained. The paper presents images and spectral data for FAEs strongly

supporting the author's hypothesis for a low energy generation mechanism with an upper energy limit between ~8–11 eV which excludes a formation caused by precipitating electrons. The authors clearly state that the exact generation mechanism remains unclear. Their finding that FAEs are associated with elevated electron temperatures points to Farley-Buneman instabilities as a potential energy source and sets an important base for follow-up studies. I have only a few minor comments for the authors to consider a few minor additions prior publication.

AR: The above statement captures the aim of the present study very well and we are grateful that the referee acknowledges the scope of the paper as a "first report" to characterise the main characteristics of FAEs.

Specific comments

Major comments: No major comments.

Minor comments:

- It would be helpful to add a video showing an example for a category 2 FAE.

AR: We agree, but unfortunately we do not have an ASK video observation of category 2 FAEs at this time.

- L. 29–30: I recommend to add references for the following papers all presenting strong arguments against the hypothesis that precipitating electrons are responsible for picket fence structures below the purple arc of STEVE (Nishimura et al., 2019). Paper 1: Gillies D. M. et al. (2019). First Observations From the TREx Spectrograph: The Optical Spectrum of STEVE and the Picket Fence Phenomena, Geophysical Research Letters, 46 (13), 7207–7213. Paper 2: Mende S. B. & Turner C. (2019). Color Ratios of Subauroral (STEVE) Arcs, Journal of Geophysical Research: Space Physics, 124 (7), 5945–5955. Paper 3: Mende S. B., et al. (2019). Subauroral Green STEVE Arcs: Evidence for Low-Energy Excitation, Geophysical Research Letters, 46 (24), 14256–14262.

AR: We will add this argumentation and suggested references 2 and 3 against precipitation-caused picket fence structures to the revised manuscript for a more balanced discussion on this point. We would like to note that the suggested reference 1 (Gillies et al., 2019) does not conclude this and rather suggests that the picket fence structures are caused by particle precipitation, with typical auroral OI emissions dominating at 557.7 nm. This paper could thus be added as an additional reference for the viewpoint that the picket fence is likely an auroral feature.

- L. 47–48: The authors mention that similar structures (FAEs) have been sighted on Svalbard at other days. I recommend to mention on how many days FAEs have been identified.

AR: FAEs were observed at least on three other dates. Since we are not able to systematically search for these features in, for example, EISCAT data or optical images yet, identification of further events is currently based on manually reviewing auroral images. One of the main goals of the present study is to derive the main characteristics of FAEs to hopefully make the identification of further events easier, as the referee has correctly pointed out.

- L. 53–56: [...The images were taken using an exposure time of 4 s and an ISO of 16000 at a cadence of 11 to 12 s, with a mean interval length of 11.8 s. This variance is due to variations of the read-out time to the attached computer, with the camera exposure time set to 10 s...] Contradicting exposure times. What is correct, 4 s or 10 s? Please clarify.

AR: We apologize for this obvious error. The exposure time is 4 s, the longer cadence of ∼11.8 s is due to a readout delay between the camera and the third-party software on the connected computer. This will be corrected in the revised manuscript.

- Figure 4: This figure shows a mark for the zenith. Is this the local magnetic zenith? Please clarify.

AR: The marked zenith is not referring to the local magnetic zenith, but rather to the geographic zenith (centre of the ASC image). This was mainly used during the analysis to derive the pixel scale of the ASC images using an equisolid projection, as the fisheye lens will result in larger pixel-to-km-ratios further from zenith, which need to be accounted for. We will add a sentence to explain this in the revised manuscript.

- Figure 6 and 7: [...Data points with errors $> 50\%$ of the values were removed...] What are the errors for the shown data points? Are they close to 50% or significantly less? Please clarify.

AR: The errors for the shown data points are mostly significantly less than 50%, further decrease of the filtering range (for example to $>30\%$) does not remove significantly more data, none of which is at the time of the FAE passing. The errors for the period between 18:20–18:23 up to the FAE passing and above 100 km (which is the most relevant part for our analysis) are less than 20% of the values. We will add a similar concise statement to the revised manuscript.

Technical corrections: None

---

## Author Comment (AC2) · 22 Oct 2020

AR: We thank the referee for providing their helpful feedback! In the following, we respond (Authors' Response, blue) to each of the referee's comments (black) individually.

In general, I disagree with the premise of the paper to create a name for an aurora-like phenomena, because the aurora in general contains so much natural variation and on a continuum of spatial and temporal scales. However, this does not invalidate the careful and thoughtful work that the authors have done in analyzing the available data for several specific auroral events. I do not want this paper rejected right out, but I would like to see less emphasis on trying to establish a new name for an aurora-like

phenomenon and more emphasis on the analysis of small-scale auroral features, which show interesting aspects when analyzed in such detail, for example multiple processes for electron acceleration happening in close proximity or even on the same field line at different places. It seems that the features discussed only happen in conjunction with aurora and are thus part of the aurora.

AR: We agree that the focus should not be on finding names for newly reported features, but since there is so much variation in the aurora it is much simpler to describe specific features when there is a named pointer associated with the particular discussed features.

However, aurora-related phenomena are not necessarily part of aurora. For instance, STEVE is not aurora, because studies have shown that it is not caused by particle precipitation. But it is sometimes accompanied by the green "picket fence" structures, which many studies have suggested to be a particle precipitation feature, and thus aurora. As pointed out by referee 1, there are also examples of studies arguing that the picket fence is not precipitation-based. Locally generated features are certainly possible in a disturbed ionosphere, such as during the analysed events, and the available data suggest a different generation mechanism than particle precipitation.

A more thorough analysis of the electron acceleration processes would absolutely be of interest, but drawing clear conclusions about these processes at such small scales is very difficult with the available data. This is outside the scope of the present study, which aims to provide a general overview of FAE characteristics and hopefully point to a potential generation mechanism that further studies might then be able to analyse in more detail.

Specific comments:

Figure 1 does not show any clear evidence of anything other than a typical auroral display. Also, the actual features cannot be seen under the yellow areas overlaid on the image.
AR: This is true. Figure 1 shows "All 262 marked FAE candidates for event 3, overlaid on the first image of the series taken at 07:36:35 UT. [...]", as mentioned in the caption. The FAEs itself did not all occur at the time of this specific picture, which is simply the first in the analysed series for that date. Our aim with this figure is to show the distribution of these features over the all-sky camera (ASC) field-of-view and illustrate the variety of shapes and sizes. We will make this point more clear in the revised manuscript, in both the running text and caption of Figure 1.

Figure 3 also does not show anything convincing either. There is no scale to gauge the size of these features extracted from the all-sky camera images. How do these features differ from what has been termed enhanced aurora (see Hallinan, et al., 1985)?

AR: We agree that a pixel/length/degree scale on the side of one of the panels would be very helpful and will add this in the revised manuscript. While the enhanced aurora (EA; Hallinan et al., 1985) describes an enhanced emission in a thin height layer along (typically) a rayed auroral structure, FAEs do not have the vertical extent of the auroral rays associated with them. The observed FAEs were also clearly dislocated from the field lines of the adjacent "normal" auroral features. As EA shows a similar spectrum to normal aurora, FAEs lack the blue emission component, at least in the samples analysed in this study. Furthermore, EA has also been observed as a quasi-stable structure lasting for minutes, while none of the FAEs lasted for that long (generally less than a minute, often only a few seconds). Overall, this suggests that they are a different phenomenon. We will add a paragraph on the comparison to EA in the revised manuscript.

Section 2.1: Inadequate description of how these are identified (just identified by eye). This will generate selection biases and their identification in general is based on some thresholds visible in the images that depend on the sensitivity of the camera and the eye. For example, if a more sensitive camera were used, it is possible that a diffuse background of aurora would become visible and these spots are just localized enhancements of that background. How are they identified, what metrics are used to determine

their boundaries, identifying them by eye is not good enough...and can easily introduce errors and biases.

AR: We agree that the methods/approach used for the identification should be described in more detail and will add this to Section 2.1 in the revised manuscript. In auroral physics visual identification is a standard approach, since there is no robust automatic auroral identification tool available. It does bring in some human-observer bias, which thus makes it important to document the selection criteria as accurately as possible. Visual thresholding is the first step to identify any auroral structures. It automatically means that we cannot claim that we have found all the features, but perhaps only the most intense ones. The manual identification process will introduce some errors, which we try to address by ordering the observations into confidence groups. As the characteristics of the higher and lower confidence groups agree well, we are confident that most candidates are observations of the same phenomenon, but concede that some other auroral features might have been falsely included in the lower confidence groups, as stated in the manuscript (line 181 ff.). The background aurora for FAEs, wherever existing, is red, and thus would not explain the observed green emission "blob" at lower altitudes.

Section 3.1: The paper mentions that the larger patches identified might just be diffuse auroral patches. Is it not possible that all of these FAEs might just belong to the general category of diffuse aurora? Pulsating / diffuse auroral patches have been found to have very limited altitude extent (see Stenbaek-Nielsen and Hallinan, 1979) and is believed to be a fairly common feature among the diffuse auroral structures.

AR: Pulsating patches do indeed have a limited altitude extent and they are very common (60% of the aurora at 3-5 MLT as estimated by Jones et al., 2011), but they occur within diffuse aurora. The analysed FAEs are seen alongside discrete arcs. Pulsating auroral patches are also typically much larger and very stable (e.g., Humberset et al., 2018; Nishimura et al., 2020) with the whole patch or a part of it undergoing quasi-periodic fluctuations in the emission intensity. However, FAEs are very shortlived (generally less than a minute, many just a few seconds) without any sign of obvi-
ous emission intensity fluctuations. The available ASK observations of FAEs also look
markedly different from diffuse aurora and pulsating patches, as they are much smaller
and show more dynamic motion. Overall, this suggests that they are a different phe-
nomenon. We will add a paragraph on the comparison to pulsating/diffuse patches in
the revised manuscript.

Line 152: Can you provide a reference that discusses the details of the electron energy
estimate from that emission line ratio?

AR: This approach is explained in, for example, Lanchester et al. (2009), see the
reference at line 158. We will add a reference also at line 152 in the revised version
of the manuscript. It is commonly used with ASK data, other references are, e.g.,
Dahlgren et al. (2016), Whiter et al. (2010), Lanchester & Gustavsson (2013).

Line 180: I would not say that it is 'clear'. The data presented seem to just show normal
variations within the diffuse aurora.

AR: We agree that the word "clear" should be avoided here and will rephrase this.
The Discussion can be started by saying that "Fragmented Aurora-like Emissions have
been studied [. . .]". We disagree that these features are just normal variations within
diffuse aurora, as they show too many specific characteristics which do not fit well to
any previously described phenomenon (see above response).

Line 184-185: It is not clear how the 'field-aligned emission extent' is measured. Paral-
lax is not used, so is it just the off zenith viewing geometry of most of the all-sky FOV?
If the latter is the case, there will likely be large uncertainty in the altitudes just based
on the viewing geometry and what other auroral features could lie along the same line
of sight at different altitudes. If it is the EISCAT signatures, then the wording should
reflect the altitude of ionization and not auroral emissions.

AR: The lack of field-aligned extent is seen in the off-zenith parts of the optical ASC

images, but the same is also suggested by the EISCAT electron density measurements of the FAEs at magnetic zenith. A precise determination of FAE altitudes is not possible with the available data, as we would either need observations from multiple locations or many more FAE signatures in EISCAT data, as discussed in line 186 ff. We will add "[...], as suggested by the off-zenith parts of the ASC images and field-aligned ionisation measured by the ESR, [...]" at line 185 in the revised manuscript.

Line 215-216: Is it not still possible that the $O^+$ density could be up to $\sim$10 times higher than the $N_2^+$ and $O_2^+$ densities in this altitude range?

AR: The $O^+$ density could be higher than the combined $O_2^+$, $NO^+$ and $N_2^+$ densities towards the upper end of this altitude range, and according to the International Reference Ionosphere (IRI) predictions for the date of event 3, $O^+$ is the dominant ion above $\sim$220 km. However, the point here is to give a lower limit of the electric field strength estimate. We cannot exclude that the IRI model underestimates the increase of molecular ions over $O^+$ towards higher altitudes that occurs at geomagnetically active times. In the most extreme case molecular ions could dominate even up to 300 km altitude. Even if we assumed $m_i = 16$ (exclusively $O^+$ ions), it would only increase the estimated $E_\perp$ value by $\sim$37%. A more reasonable guess would be closer to $m_i = 22$ (considering the very low $O^+$ abundances at the lower altitudes), which would result in a $\sim$17% increase of $E_\perp$. We will reformulate the statement to make it clearer that the estimated $E_\perp \approx 70$ mV m$^{-1}$ is a lower limit (if molecular ions are not dominating). In any case, a higher value would only further exceed the typical threshold for Farley-Buneman instabilities and thus not change the argumentation/conclusion for this point.

Lines 253-257: This paragraph summarizes the overall uncertainty and limitations of the data used in the current study, which limits the conclusions that can be drawn from them. Thus it is not scientifically sound to define a new feature with such a limited data set, especially without any clear metrics of how they are defined. Multitudes of auroral structures have been observed within both discrete and diffuse aurora for many

decades and only very few have been assigned specific names and those come mostly from historical reasons.

AR: It is important to be clear and honest about the limitations in the identification. However, hundreds of observations is not an insignificant number of features which do not fit into any earlier reported type of aurora. We will extend Section 2.1 to improve the metrics description and more clearly state how the events have been identified. As discussed above, visual identification of auroral features is a standard approach, and the aim of the present study is not to present a definite answer on the nature of FAEs or their origin, but rather to provide a first overview of their apparent characteristics based on the available data. Based on our analysis, we arrive at the conclusion that the observed features do not fit the characteristics of any previously reported auroral features, which is why we present them as a potential newly reported phenomenon. We disagree with the idea that features should not be named just because many other features do not have names yet. Naming and reporting on a rare and unexplained phenomenon is one of the first steps to gathering further observations and thus ultimately to understanding its formation mechanism.
* * *
References:

Hallinan, T. J., H. C. Stenbaek-Nielsen, and C. S. Deehr (1985), Enhanced aurora, J. Geophys. Res., 90, 8461–8475, doi:10.1029/JA090iA09p08461.

Stenbaek-Nielsen, H. C. and T. J. Hallinan (1979), Pulsating Auroras: Evidence for Non-collisional Thermalization of Precipitating Electrons, J. Geophys. Res., 84, 32573271.

Response references:

Jones, S. L., Lessard, M. R., Rychert, K., Spanswick, E., and Donovan, E. (2011), Large-scale aspects and temporal evolution of pulsating aurora, J. Geophys. Res.,

116, A03214, doi:10.1029/2010JA015840.

Humberset, B. K., Gjerloev, J. W., Mann, I. R., Michell, R. G., & Samara, M. (2018). On the Persistent Shape and Coherence of Pulsating Auroral Patches. Journal of Geophysical Research: Space Physics, 123(5), 4272–4289. https://doi.org/10.1029/2017JA024405

Nishimura, Y., Lessard, M. R., Katoh, Y., Miyoshi, Y., Grono, E., Partamies, N., Sivadas, N., Hosokawa, K., Fukizawa, M., Samara, M., Michell, R. G., Kataoka, R., Sakanoi, T., Whiter, D. K., Oyama, S. ichiro, Ogawa, Y., & Kurita, S. (2020). Diffuse and Pulsating Aurora. Space Science Reviews, 216(1), 1–38. https://doi.org/10.1007/s11214-019-0629-3

Dahlgren, H., Lanchester, B. S., Ivchenko, N., & Whiter, D. K. (2016). Electrodynamics and energy characteristics of aurora at high resolution by optical methods. Journal of Geophysical Research: Space Physics, 121(6), 5966–5974. https://doi.org/10.1002/2016JA022446

Whiter, D. K., Lanchester, B. S., Gustavsson, B., Ivchenko, N., & Dahlgren, H. (2010). Using multispectral optical observations to identify the acceleration mechanism responsible for flickering aurora. Journal of Geophysical Research: Space Physics, 115(12), 1–10. https://doi.org/10.1029/2010JA015805

Lanchester, B. and Gustavsson, B. (2013). Imaging of Aurora to Estimate the Energy and Flux of Electron Precipitation. In Auroral Phenomenology and Magnetospheric Processes: Earth And Other Planets (eds A. Keiling, E. Donovan, F. Bagenal and T. Karlsson). doi:10.1029/2011GM001161